# Investigating the Role of GDF-15 in Diabetes and Obesity: A Comprehensive Analysis of a Cohort from the KDEP Study

**DOI:** 10.3390/biomedicines13071589

**Published:** 2025-06-30

**Authors:** Jehad Abubaker, Mohamed Abu-Farha, Ahmed N. Albatineh, Irina Al-Khairi, Preethi Cherian, Hamad Ali, Ibrahim Taher, Fahad Alajmi, Mohammed Qaddoumi, Muhammad Abdul-Ghani, Fahd Al-Mulla

**Affiliations:** 1Department of Biochemistry and Molecular Biology, Dasman Diabetes Institute, Dasman 15462, Kuwait; mohamed.abufarha@dasmaninstitute.org (M.A.-F.); irina.alkhairi@dasmaninstitute.org (I.A.-K.); preethi.cherian@dasmaninstitute.org (P.C.); muhammad.abdulghani@dasmaninstitute.org (M.A.-G.); 2Department of Translational Research, Dasman Diabetes Institute, Dasman 15462, Kuwait; fahad.alajmi@dasmaninstitute.org; 3Department of Biostatistics and Health Data Science, College of Health, Lehigh University, Bethlehem, PA 18015, USA; aha424@lehigh.edu; 4Department of Medical Laboratory Sciences, Faculty of Allied Health Sciences, Health Sciences Center (HSC), Kuwait University, Jabriya, P.O. Box 24923, Safat 13110, Kuwait; hamad.ali@ku.edu.kw; 5Microbiology Unit, Department of Pathology, College of Medicine, Jouf University, Sakaka P.O. Box 2014, Saudi Arabia; itaher@ju.edu.sa; 6Pharmacology and Therapeutics Department, Faculty of Pharmacy, Kuwait University, Kuwait City 13110, Kuwait; mohammad.qaddoumi@dasmaninstitute.org; 7Division of Diabetes, University of Texas Health Science Center, San Antonio, TX 78229, USA

**Keywords:** GDF-15, diabetes, obesity, insulin resistance, ethnicity

## Abstract

**Background:** Growth differentiation factor 15 (GDF-15), a member of the transforming growth factor-β (TGF-β) superfamily, is upregulated under cellular stress conditions and has emerged as a potential biomarker for metabolic disorders. However, its expression in relation to diabetes and obesity across different demographic groups remains understudied. This study investigated the association between plasma GDF-15 levels, diabetes mellitus, and obesity in individuals of varying ages, ethnicities, and genders. **Methods:** In a cross-sectional study, plasma GDF-15 concentrations were measured in 2083 participants enrolled in the Kuwait Diabetes Epidemiology Program (KDEP). The dataset included anthropometric, clinical, biochemical, and glycemic markers. Multivariate regression analysis was used to examine associations between GDF-15 levels and metabolic phenotypes. **Results:** Mean plasma GDF-15 levels were significantly higher in males than females (580.6 vs. 519.3 ng/L, *p* < 0.001), and in participants >50 years compared to those <50 years (781.4 vs. 563.4 ng/L, *p* < 0.001). Arab participants had higher GDF-15 levels than South and Southeast Asians (597.0 vs. 514.9 and 509.9 ng/L, respectively; *p* < 0.001). Positive correlations were found with BMI, waist and hip circumferences, blood pressure, insulin, and triglycerides; negative correlations were observed with HDL cholesterol. Median regression indicated that elevated GDF-15 levels were independently and significantly associated with male gender, older age, obesity, diabetes, and insulin resistance. Adjusted median regression indicated that male gender (β = 30.1, 95%CI: 11.7, 48.5), older age (β = 9.4, 95%CI: 8.0, 10.7), and insulin resistance (β = 7.73, 95%CI: 1.47, 14.0) indicated a significant positive association with GDF-15. South Asian participants (β= −41.7, 95%CI: −67.2, −16.2) had significantly but Southeast Asian participants (β= −23.3, 95%CI: −49.2, 2.56) had marginally significantly lower GDF-15 levels compared to participants of Arab ethnicity. **Conclusions:** Higher GDF-15 levels are associated with age, male gender, Arab ethnicity, obesity, and diabetic traits. These findings support the potential role of GDF-15 as a biomarker for metabolic disorders, particularly in high-risk demographic subgroups.

## 1. Introduction

Obesity and diabetes are global health problems with a significant burden on the world economy and are responsible for numerous health-related problems worldwide [1]. The primary cause of obesity is multifactorial and results from a complex interplay of genetic, environmental, and behavioral factors. The interaction among these factors contributes to the complexity of obesity and poses challenges for its treatment, given the intricate relationships among various genes and other risk factors such as environmental and lifestyle factors [2]. While an individual’s genetic background is one of the essential factors contributing to obesity, it is important to note that the basis of obesity is not solely genetic [3]. Evidence suggests that genes often need to be closely linked with environmental and lifestyle risk factors to impact weight [4]. Therefore, gaining a deeper understanding of the common causes of obesity and weight gain is crucial. Lifestyle interventions such as physical activity and diet represent the initial therapeutic approach for managing obesity and diabetes. However, there is an urgent need for novel therapies.

Growth differentiation factor 15 (GDF-15) is a member of the transforming growth factor-β (TGF-β) superfamily and is highly expressed in several tissues, including the liver, heart, intestine, and kidneys [5,6,7]. It was primarily identified as a soluble factor produced by macrophages and cancer cells. GDF-15 is induced in all cell types in response to mitochondrial and endoplasmic reticulum stress, which is commonly observed in various conditions such as cancer, diabetes, inflammation, and chronic liver and kidney disease [8]. Increased body weight and adipose tissue mass are also positively associated with elevated GDF-15 expression and release [9]. Over the past decade, many investigations have emphasized the importance of GDF-15 in several diseases. These investigations demonstrated the significant clinical relevance of GDF-15 as a diagnostic and prognostic biomarker for conditions including prostate cancer, pulmonary disease, diabetic cardiomyopathy, heart failure, and mitochondrial disease [10]. Notably, a study utilizing human liver single-cell RNA sequencing has shown increased expression of GDF-15 across all hepatocytes, which supports the strong correlation between GDF-15 levels and non-alcoholic fatty liver disease (NAFLD) and obesity [11]. Several regulatory pathways have been proposed to explain the increased expression and secretion of GDF-15. For instance, the stress-responsive transcription factor p53 and early growth response factor-1 (EGR1) have been described as key regulators of GDF-15 [12]. Moreover, the activation of AMP-activated protein kinase (AMPK) increases GDF-15 expression, and its transcription requires the activating transcription factor 4 (ATF4) and C/EBP homologous protein (CHOP) [13]. Both ATF4 and CHOP are activated by mitochondrial stress and are commonly associated with obesity and insulin resistance [9,14].

In animal studies, GDF-15 promotes weight loss by reducing fat mass and adiposity, improving insulin sensitivity and glucose tolerance while maintaining energy expenditure [8]. However, there has also been a claim that GDF-15 does not change energy expenditure but suppresses food intake through its direct effect on the area postrema and nucleus tractus, which are highly enriched in the glial cell-derived neurotrophic factor (GDNF) family receptor α-like (GFRAL), known for its high affinity to GDF-15 [15]. Although the exact region of the brain that contributes to the reduction in food intake is not fully understood, several mechanistic pathways, including food preferences, nausea, vomiting, delayed gastric emptying, and taste aversion, were proposed [12,16]. These proposed mechanistic pathways appear to be independent of the effects of leptin and glucagon-like peptide 1 (GLP-1) [16,17].

Obesity is a risk factor for type 2 diabetes mellitus (T2DM), and some evidence from animal studies suggests that treatment with GDF-15 decreases body weight and glucose levels, primarily due to the loss of fat mass and reduced food intake [17]. However, a recent study proposed that GDF-15 decreases glucose levels independently of changes in body weight but through stimulating insulin secretion [18]. On the other hand, chronic low-grade inflammation is a culprit for insulin resistance and other metabolic derangements. Several studies have shown that GDF-15 may improve glucose homeostasis by enhancing insulin secretion and reducing low-grade chronic inflammation [19]. Considerable data indicate that metformin, a commonly prescribed medication for diabetes management, is associated with increased GDF-15 stimulation, subsequently improving glycemic control and inducing weight loss [20]. A similar effect has been observed with high- and moderate-intensity exercise [21,22]. Non-alcoholic fatty liver disease (NAFLD), which reflects the degree of obesity and the risk for T2DM, is associated with increased expression and release of GDF-15, making it a potential biomarker for NAFLD and disease progression, particularly for worsening fibrosis and steatosis [23].

Recognizing the pivotal role of GDF-15 in metabolic disorders, it becomes imperative to explore its relationship with variables such as gender and ethnicity, particularly in regions where obesity and diabetes are prevalent. This study is designed to investigate the levels of GDF-15 in individuals afflicted with diabetes, obesity, and insulin resistance. Additionally, it seeks to identify specific metabolic characteristics linked to variations in GDF-15 levels. By focusing on these aspects, this research addresses a critical and specialized area of interest, shedding light on the intricate interplay between GDF-15 and prevalent metabolic conditions.

## 2. Materials and Methods

### 2.1. Participants and Study Design

The current study is a cross-sectional analysis of data from the Kuwait Diabetes Epidemiological Program (KDEP), which recruited a representative sample of adults (aged ≥ 18 years) in Kuwait from 2011 to 2014. This study received approval from the Ethical Review Committee of Dasman Diabetes Institute (Protocol number RA-2010-004) and was conducted in accordance with the ethical standards of the Declaration of Helsinki. All participants provided written informed consent before they participated in this study. Random sampling of the Kuwaiti population with proportional representation from each of the seven governorates was conducted for participant recruitment. A list of Kuwaiti residents, complete with their unique identification codes, was provided by the National Public Authority of Civil Information. A stratified random sampling technique was employed to select survey participants from this resident list. The survey design was adapted from the WHO STEPwise approach to surveillance (STEPS) methodology [24]. Individuals suffering from any infection and those aged younger than 18 or older than 65 were excluded from recruitment. Recruitment took place at the Dasman Diabetes Institute between April 2011 and June 2014, with a dedicated team consisting of nurses, coordinators, interviewers, and phlebotomists.

### 2.2. Anthropometry and Vital Signs Measurements

Anthropometric measurements, including body weight, height, and waist circumference (WC), as well as vital signs such as blood pressure (BP), were recorded for each participant. BP was assessed using an Omron HEM-907XL Digital Sphygmomanometer. Three systolic and diastolic blood pressure readings were taken with 5–10 min of rest between each reading, and the average values of the systolic and diastolic blood pressure readings were recorded. Height and weight were measured while the participants were dressed in lightweight indoor clothing and were barefoot, utilizing calibrated portable electronic weighing scales and portable inflexible height measuring bars. WC was measured at the highest point of the iliac crest and the mid-axillary line using a constant tension tape after a normal exhalation, with the arms in a relaxed position at the sides. The body mass index (BMI) was calculated using the standard formula as follows: body weight (in kilograms) divided by the square of height (in meters).

### 2.3. Laboratory Measurements

Blood samples were collected after confirming that the participants had fasted for at least 10 h overnight. Blood samples were drawn into Vacutainer EDTA aprotinin tubes. Plasma was obtained after centrifugation for 10 min at 2000× *g* at room temperature. Subsequently, plasma was aliquoted into cryogenic tubes and stored at −80 °C. Prior to analysis, plasma samples stored in freezers at −80 °C were thawed on ice and then centrifuged at 10,000× *g* for 5 min at 4 °C to remove any debris. Blood samples were used to measure lipid and glycemic profiles, including fasting plasma glucose (FPG), hemoglobin A1c (HbA1c), fasting insulin, triglycerides (TG), total cholesterol (TC), low-density lipoprotein (LDL), and high-density lipoprotein (HDL). Glucose and lipid profiles were assessed using the Siemens Dimension RXL chemistry analyzer (Diamond Diagnostics, Holliston, MA, USA), whereas HbA1c levels were determined using the VariantTM device (BioRad, Hercules, CA, USA). All laboratory assessments were conducted by certified technicians at the clinical laboratories of DDI, following approved methods and quality standards established by the Ministry of Health. Insulin levels were quantified using the Access Ultrasensitive Insulin Assay (Beckman Coulter, Brea, CA, USA), with both intra- and inter-assay coefficients of variation not exceeding 6%. Insulin resistance was calculated using the Homeostatic Model Assessment for Insulin Resistance (HOMA-IR) formula: (FBG in mmol/L) × (fasting insulin in mU/L)/22.5.

### 2.4. Definition of Prediabetes and Diabetes

The WHO criteria for the diagnosis of diabetes were used: FPG ≥ 7 mmol/L or HbA1c ≥ 6.5% (48 mmol/mol) [25]. The participants were considered to have diabetes if they met one of the following criteria: (1) self-reported a previous diagnosis of diabetes with concurrent prescription of glucose-lowering medication; (2) recorded a survey-measured FPG ≥7 mmol/L; or (3) recorded a survey-measured HbA1c ≥6.5%. Diabetes unawareness was determined if the survey measured elevated FPG or HbA1c without a prior self-reported diagnosis. The participants who self-reported a previous diabetes diagnosis but were not currently receiving treatment for diabetes and recorded a survey-measured FPG <7 mmol/L and a survey-measured HbA1c <6.5% were considered “unverified” (*n* = 13). These individuals were reclassified as having normal glycemia or prediabetes if they met the prediabetes criteria. For prediabetes, the WHO criteria for FPG were used, and the International Expert Committee (IEC) criteria for HbA1c were used [25]. The participants were considered to have prediabetes if they did not meet any of the criteria for the diagnosis of diabetes and recorded a survey-measured FPG in the range of 6.1–6.9 mmol/L or a survey-measured HbA1c in the range of 6.0–6.4% (42–47 mmol/mol). Normal glycemia was defined as FPG ≤ 6 mmol/L and HbA1c ≤ 5.9% (≤41 mmol/mol) among individuals without diabetes. Impaired fasting glucose (IFG) was defined by fasting blood glucose values ≥ 5.6 and <7 mmol/L.

### 2.5. GDF15 Plasma Levels and R&D Custom Multiplexing Assay

Plasma samples were extracted, aliquoted into plates, and stored at −80 °C for future use. For the multiplexing analysis, plasma samples were thawed and diluted 2X following the kit instructions for the Luminex custom-made panel (cat #LXSAHM, R&D, CA, USA). The procedure was performed according to the kit instructions. In summary, plasma samples were diluted with the sample buffer provided in the kit. The kit standard was prepared with a 3-fold serial dilution. A cocktail of antibodies complexed to magnetic beads was diluted and aliquoted into a 96-well plate. The samples and standards were then transferred onto the plate and incubated for 2 h. After the incubation, the plates were washed and incubated in a diluted biotinylated antibody cocktail for 1 h. This was followed by a washing step and incubation with diluted streptavidin-PE for 30 min. Before obtaining the results, the plates were washed, and the beads were resuspended with the assay buffer provided with the kit. Data were acquired using the Bioplex-200 (Bio-Rad, CA, USA) according to the kit specifications, and the results were calculated using a 5-PL nonlinear standard curve setting in the Bio-Plex manager software version 6.0. No significant cross-reactivity with other proteins was observed. The intra-assay coefficients of variation ranged from 1.2% to 3.8%, whereas the inter-assay coefficients of variation ranged from 6.8% to 10.2%.

### 2.6. Statistical Analysis

The statistical software STATA version 14 (STATA Corp) was used for data analysis. Initially, the data were examined for abnormalities and then re-coded as necessary. Continuous variables were presented as mean (SD) if the normality assumption was met; otherwise, the median (IQR) was reported. To assess significant differences between a continuous covariate dichotomized over a binary variable, the two-sample *t*-test was employed if the normality assumption for both groups was satisfied; otherwise, the Mann–Whitney U test was used. Differences in outcomes over a categorical exposure were evaluated using ANOVA if normality and homogeneity of variances were met; otherwise, the Kruskal–Wallis test was employed. To measure the strength of the correlation between binary and continuous variables, the point-biserial correlation coefficient was calculated. For two continuous covariates, the Pearson correlation coefficient was computed if normality was observed for both variables; otherwise, the Spearman rank correlation coefficient was used. To model the relationship between a continuous outcome and a set of covariates in cases where the distribution of the outcome was skewed and outliers were present, quantile (median) regression was employed. This method is known for its robustness against outliers and its ability to handle over-dispersion or under-dispersion. In addition, the estimated robust variance–covariance matrix of the estimators (VCE) was obtained through bootstrapping. All statistical tests were two-tailed, and the significance level was set at 5%.

## 3. Results

### 3.1. Study Sample

The final sample size included in the analysis was *n* = 2083. Descriptive analysis indicated that the majority of patients were male (55.7%), with a median age of 45 years (min = 18, max = 82, IQR = 16), and 36% were less than 40 years old. Additionally, 46.6% identified as being of Arab ethnicity, 30.8% had type 2 diabetes mellitus (T2DM), 40.2% were overweight, and 38.7% were obese. Furthermore, based on the HOMA-IR score, 49.7% were insulin-resistant, with more details provided in Table 1.

The average GDF-15 concentration across the sample was significantly higher in males than in females (580.6 (308.3) ng/L vs. 519.3 (265.0) ng/L, *p* < 0.001, respectively, Figure 1a), and in those who were >50 years old than those <40 years of age (781.4 (516.6) ng/L vs. 471.5 (171.8) ng/L, *p* < 0.001, respectively, Figure 1c). Moreover, participants from Arab ethnic backgrounds had higher GDF-15 levels than South Asian and Southeast Asian participants (597.0 (367.5) vs. 514.9 (236.5) and 509.9 (198.0) ng/L, *p* < 0.001, respectively, Figure 1b). Furthermore, obese individuals and patients with diabetes had significantly higher GDF-15 levels compared to others (see Table 2). A significant difference was observed in the median level of GDF-15 between normal-weight and obese participants (502.9 (218.5) ng/mL vs. 598.1 (346.2) ng/mL, *p* < 0.001, Figure 2a).

T2DM patients have significantly higher median GDF-15 (839.9 (625.8) ng/mL) compared to non-diabetic ones (528.2 (244.4) ng/mL), with *p* < 0.001 (Figure 2b). Finally, a significant difference was observed with increased insulin resistance (HOMA-IR score: Normal ≤ 2 = 520.6 (231.3) ng/mL, and IR > 2 = 597.4 (351.4) ng/mL) (Figure 2c).

### 3.2. Correlation Between GDF-15 and the Clinical Markers

In the multivariate analyses, it was observed that GDF-15 levels exhibited statistically significant positive associations with several factors, including age (r = 0.528, *p* < 0.001), BMI (r = 0.181, *p* < 0.001), hip circumference (r = 0.159, *p* < 0.001), waist circumference (WC) (r = 0.276, *p* < 0.001), systolic blood pressure (SBP) (r = 0.230, *p* < 0.001), and diastolic blood pressure (DBP) (r = 0.105, *p* < 0.001). In addition, GDF-15 levels displayed a positive association with triglyceride (TG) (r = 0.161, *p* < 0.001) levels but a negative association with high-density lipoprotein (HDL) levels (r = −0.118, *p* < 0.001, Table 3).

### 3.3. Correlation Between GDF-15 and the Glycemic Indices

The analysis (Table 3) revealed a significant and positive association between plasma GDF-15 levels and HbA1c (r = 0.297, *p* < 0.001), fasting blood glucose (FBG) (r = 0.269, *p* < 0.001), fasting insulin (r = 0.127, *p* < 0.001), and the homeostatic model of insulin resistance (HOMA-IR) (r = 0.198, *p* < 0.001). Although some bivariate correlations between GDF-15 and clinical variables such as BMI, insulin, and blood pressure reached statistical significance, the effect sizes were small. These findings, while statistically detectable due to the large sample size, may have limited standalone clinical relevance. However, they served as a rationale for including these covariates in the quantile regression model, which provides a more comprehensive understanding of the relationship between GDF-15 and metabolic risk factors.

### 3.4. Correlation Between GDF-15 and the Biochemical Markers

GDF-15 exhibited significant and positive associations with alanine transaminase (ALT) (r = 0.113, *p* <0.001), c-reactive protein (CRP) (r = 0.106, *p* < 0.001), creatinine (r = 0.209, *p* <0.001), FGF-23 (r = 0.160, *p* < 0.001), RAGE/AGER (r = 0.107, *p* < 0.001), CXCL10/IP10 (r = 0.111, *p* < 0.001) and white blood cell (WBC) levels (r = 0.098, *p* <0.001), as indicated in Table 3.

In a secondary analysis using quantile median regression, and after adjusting for gender, age, ethnicity, diabetes status, HbA1c, HOMA-IR, and BMI, certain associations remained robust. Specifically, there was still a strong positive association between GDF-15 levels and male gender (β: 9.4, 95% CI: 8.0, 10.7, *p* < 0.001) as well as age (β: 9.4, 95% CI: 8.0, 10.7, *p* < 0.001). Conversely, a negative association was observed between GDF-15 levels and individuals of South Asian ethnic background (β: −41.7, 95% CI: −67.2, −16.2, *p* < 0.001), as detailed in Table 4.

## 4. Discussion

This cross-sectional study revealed a compelling link between circulating GDF-15 levels and several crucial clinical, biochemical, and glycemic markers within the KDEP study cohort. Notably, our research demonstrates that elevated GDF-15 levels are closely associated with an increased risk of metabolic complications and heightened inflammation in individuals affected by both obesity and diabetes. These observed positive correlations between the GDF-15 levels and obesity/diabetes markers appear to be a compensatory response to the underlying pathological effects of the metabolic disruptions observed in obese and diabetic patients. In essence, our study suggests that GDF-15 may play a compensatory role in response to the adverse metabolic changes observed in obese and diabetic individuals, potentially indicating its involvement in the body’s response to these conditions. These findings could have significant implications for further research into the mechanisms underlying diabetes and obesity, as well as potential therapeutic interventions aimed at modulating GDF-15 levels to mitigate metabolic risks.

In accordance with a previous study [26], our research identifies a robust correlation between elevated GDF-15 concentrations and older age, with this association being particularly pronounced among individuals aged over 50. This strong connection to older age may be attributed to the presence of subclinical cardiovascular disease within this specific population cohort. It is plausible that GDF-15 levels rise as a response to the underlying cardiovascular pathology associated with aging. Additionally, as individuals age, there is typically a progressive decline in renal function, evident in increased creatinine levels, which, in turn, correlates with higher GDF-15 levels. This connection between GDF-15 and creatinine levels is significant, as several studies have demonstrated GDF-15’s potential to predict declines in estimated glomerular filtration rate (eGFR) and mortality in patients with type 1 diabetes mellitus (T1DM) and with nephropathy [27,28,29]. Our research also revealed notable gender differences in GDF-15 levels, with males exhibiting significantly higher concentrations. This observation aligns with prior studies that have consistently reported gender disparities in GDF-15 levels [30,31]. However, it is important to acknowledge that these gender differences may be influenced by historically higher cardiovascular risk among men. High blood pressure, elevated C-reactive protein (CRP), and increased waist circumference are recognized risk factors and predictors of cardiovascular disorders. In our study, we observed a strong association between higher GDF-15 levels and cardiometabolic risk factors, including high triglycerides (TG), low high-density lipoprotein (HDL), and hypertension. These associations are notable because they precede the onset of overt cardiovascular disorders. Our findings align with previous studies that have shown the upregulation of GDF-15 following various cardiovascular events characterized by inflammation, oxidative stress, and pressure overload [32,33].

Moreover, our study identifies a significant association between increased GDF-15 concentration and insulin resistance, as indicated by the homeostatic model assessment of insulin resistance (HOMA-IR) and serum insulin levels. Higher GDF-15 levels are also linked to increased liver enzyme levels, a biomarker of hepatic inflammation. Similar findings have been observed in patients with chronic liver disease [34,35]. Additionally, we observed significant racial differences in circulating GDF-15 levels, particularly among the South Asian cohort, prompting the need for more inclusive studies as well as genetic exploration. These future studies are essential to fully establish the clinical utility of GDF-15, paving the way for its potential applications in the realm of metabolic disorders. A result worth noting is that after adjusting for covariates in Table 4, a significant interaction was observed between diabetes status and ethnicity in relation to GDF-15 levels. This indicates that the association between diabetes and GDF-15 levels varies by ethnicity. Notably, diabetic Southeast Asian participants had significantly lower median GDF-15 levels compared to those of Arab ethnicity.

Several limitations to our study must be acknowledged. The cross-sectional nature of this study restricts our ability to infer causative effects, suggesting that GDF-15 may function as both a contributor to and a biomarker for cardiometabolic disorders. Genetic factors may strongly influence circulating GDF-15 levels, warranting future investigations into different genetic variants that may play a role in GDF-15 regulation. Given the cross-sectional nature of this study, the associations observed between GDF-15 and various metabolic markers should be interpreted as exploratory and hypothesis-generating rather than indicative of causal relationships. Future longitudinal and interventional studies are warranted to validate these findings and further elucidate the potential role of GDF-15 in metabolic disease progression.

Our study represents a substantial stride in advancing our understanding of the potential significance of GDF-15 in obesity and diabetes, highlighting its promising role as a biomarker for metabolic disease among various ages, genders, and ethnic groups. Subsequent research and validation studies hold the key to establishing the clinical relevance of GDF-15 in metabolic derangement, which could ultimately enhance our ability to manage and address these pressing health concerns more effectively.

## 5. Conclusions

In summary, our study provides a comprehensive examination of the correlation between circulating GDF-15 levels and multiple factors, including age, gender, race, diabetes, and obesity. We have effectively demonstrated a significant association between GDF-15 levels and crucial cardiometabolic risk factors, particularly insulin resistance. Nevertheless, it is noteworthy that age, gender, and ethnicity exerted a more substantial influence on GDF-15 levels. These findings enhance our understanding of GDF-15’s prospective role as a marker of diabetes and obesity amongst specific genders, age groups, and ethnicity. It is also the case that this study opens up a new horizon for future research to unearth the reason why the association is so strong with age, gender, and ethnicity.

## Figures and Tables

**Figure 1 biomedicines-13-01589-f001:**
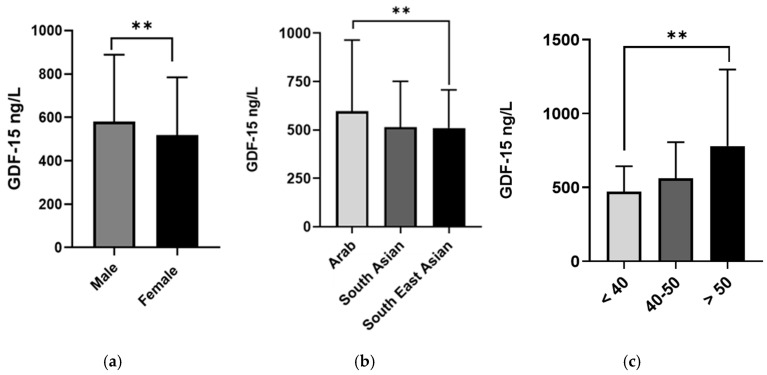
GDF-15 level in plasma in all population (*n* = 2083) categorized by gender, ethnicity, and Age. The population was stratified based on gender (male and female) (**a**), ethnicity (Arab, South Asian, and Southeast Asian) (**b**), and age (>40, 40–50, and >50 years) (**c**). The level of GDF-15 in plasma was determined using a multiplex metabolic panel. Statistical assessment was two-sided and considered statistically significant at ** *p* < 0.001.

**Figure 2 biomedicines-13-01589-f002:**
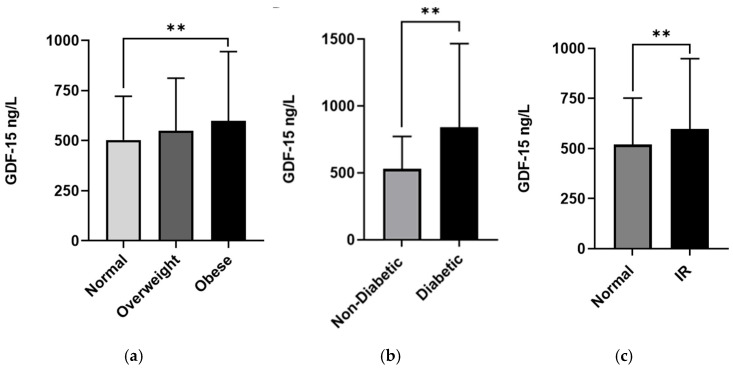
GDF-15 level in plasma in all populations (*n* = 2083) categorized by metabolic parameters. The population was stratified based on BMI (BMI: >24.99 (Normal), 25–29.9 (Overweight), ≥30 (Obese)) (**a**), diabetes status (non-diabetic and T2DM) (**b**), and insulin resistance (HOMA score: ≤2 (Normal) and >2 (IR) (**c**). The level of GDF-15 in plasma was determined using a multiplex bone panel. Statistical assessment was two-sided and considered statistically significant at ** *p* < 0.01.

**Table 1 biomedicines-13-01589-t001:** Demographic characteristics of 2083 participants.

Characteristics	Number (%)
**Age**	
<40	750 (36.0)
40–50	680 (32.7)
>50	653 (31.3)
**Ethnicity**	
Arab	899 (46.6)
South Asian	666 (34.5)
Southeast Asian	364 (18.9)
**Diabetic Status**	
Non-Diabetic	1425 (69.2)
Diabetic	633 (30.8)
**BMI**	
Normal	441 (21.2)
Overweight	837 (40.2)
Obese	805 (38.6)
**HOMA-IR**	
HOMA-IR ≤ 2	969 (50.3)
HOMA-IR > 2	958 (49.7)
Hip Circumference (HC), median (IQR)	102.3 (13)
Waist Circumference (WC), median (IQR)	95 (15)
SBP, median (IQR)	131 (26)
DBP, median (IQR)	80 (16)
HbA1c, median (IQR)	5.8 (1.2)
TC, median (IQR)	5.1 (1.33)
AST, median (IQR)	21 (8)
CRP, median (IQR)	3 (2)
Creatinine, median (IQR)	76 (24)
Vitamin D, median (IQR)	12.01 (10.8)
FBG, median (IQR)	5.3 (1.7)
Insulin, median (IQR)	7.9 (6.7)
TSH, median (IQR)	1.53 (1.14)
FT4, median (IQR)	11.78 (3.43)
FT3, median (IQR)	4.76 (0.78)
ALT, median (IQR)	37 (19)

**HOMA-IR:** homeostatic model of insulin resistance, **BMI:** body mass index, **IQR:** interquartile range, **CRP**: c-reactive protein, **FBG**: fasting blood glucose.

**Table 2 biomedicines-13-01589-t002:** Descriptive analysis of GDF-15 distribution across 2083 participants.

Characteristics	Number of Patients	GDF-15 Levels (ng/L)	*p*-Value
**Gender**			<0.001 ^a^
Male	923	580.6 (308.3)	
Female	762	519.3 (265.0)	
**Age**			<0.001 ^b^
<40	691	471.5 (171.8)	
40–50	565	563.4 (243.1)	
>50	429	781.4 (516.6)	
**Ethnicity**			<0.001 ^b^
Arab	714	597.0 (367.5)	
South Asian	511	514.9 (236.5)	
Southeast Asian	311	509.9 (198.0)	
**Diabetes Status**			<0.001 ^a^
Non-Diabetic	1405	528.2 (244.4)	
Diabetic	259	839.9 (625.8)	
**BMI**			<0.001 ^b^
Normal	389	502.9 (218.5)	
Overweight	690	549.9 (261.9)	
Obese	606	598.1 (346.2)	
**HOMA-IR**			<0.001 ^a^
HOMA-IR ≤ 2	892	520.6 (231.3)	
HOMA-IR > 2	642	597.4 (351.4)	

^a^ based on the Mann–Whitney U test, ^b^ based on the Kruskal–Wallis test. BMI: body mass index. HOMA-IR: homeostatic model of insulin resistance.

**Table 3 biomedicines-13-01589-t003:** Correlations between GDF-15 and the clinical, glycemic, and biochemical markers in the 2083 patients.

Marker	GDF-15(r)	*p*-Value	Marker	GDF-15(r)	*p*-Value	Marker	GDF-15(r)	*p*-Value
Gender ^a^	0.027	0.260	HbA1c ^b^	0.297	**<0.001**	ALT ^b^	0.113	**<0.001**
Nationality ^a^	0.108	**<0.001**	Insulin ^b^	0.127	**<0.001**	AST ^b^	0.094	**0.002**
Age ^b^	0.528	**<0.001**	TC ^b^	−0.032	<0.192	CRP ^b^	0.106	**<0.001**
BMI ^b^	0.181	**<0.001**	TG ^b^	0.161	**<0.001**	Creatinine ^b^	0.209	**<0.001**
Hip ^b^ Circumference	0.159	**<0.001**	HDL ^b^	−0.118	**<0.001**	Vitamin D ^b^	0.063	**0.014**
Waist ^b^ Circumference	0.276	**<0.001**	LDL ^b^	−0.063	**0.011**	TSH ^b^	0.007	**<0.001**
SBP ^b^	0.230	**<0.001**	TNFAα ^b^	−0.0172	0.480	FT4 ^b^	0.008	0.444
DBP ^b^	0.105	**<0.001**	FGF1 ^b^	0.0363	0.136	FT3 ^b^	0.001	0.318
FBG ^b^	0.269	**<0.001**	HOMA-IR ^b^	0.198	**<0.001**	RAGE/AGER ^b^	0.107	**<0.001**
CXCL10/IP10 ^b^	0.110	**<0.001**	FGF-23 ^b^	0.162	**<0.001**	RANTES ^b^	−0.0218	0.903

^a^ based on the point-biserial correlation, ^b^ based on the Spearman correlation coefficient. **LDL**: low-density lipoprotein, **SBP**: systolic blood pressure, **DBP**: diastolic blood pressure. The *p*-value was significant at <0.001.

**Table 4 biomedicines-13-01589-t004:** The adjusted analysis for the association between GDF-15 and the clinical, glycemic, and biochemical markers in 2083 patients.

Variable	Unadjusted Median Regression β (95%CI)	*p*-Value	Adjusted Median Regression β (95%CI)	*p*-Value
**Male gender**	61.1 (37.3, 84.8)	**<0.001**	30.1 (11.7, 48.5)	**<0.001**
**Age**	11.2 (10.1, 12.3)	**<0.001**	9.4 (8.0, 10.7)	**<0.001**
**South Asian**	−82.8 (−108.1, −57.6)	**<0.001**	−41.7 (−67.2, −16.2)	**<0.001**
**Southeast Asian**	−87.8 (−114.5, −61.1)	**<0.001**	−23.3 (−49.2, 2.56)	0.077
**Diabetic**	311.6 (209.7, 413.6)	**<0.001**	59.6 (−42.8, 161.9)	0.254
**BMI**	7.5 (5.1, 9.9)	**<0.001**	−0.49 (−2.56, 1.57)	0.641
**HbA1c**	70.6 (53.0, 88.1)	**<0.001**	14.8 (−16.1, 45.7)	0.348
**HOMA-IR**	26.8 (14.35, 39.2)	**<0.001**	7.73 (1.47, 14.0)	**0.015**

**BMI**: body mass index, **HOMA-IR:** homeostatic model of insulin resistance.

## Data Availability

The datasets used and/or analyzed during this study are available from the corresponding author on reasonable request.

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
