# Peer review of "Investigating the Role of GDF-15 in Diabetes and Obesity: A Comprehensive Analysis of a Cohort from the KDEP Study"

_biomedicines, 2025, doi:10.3390/biomedicines13071589_

Round 1

Reviewer 1 Report

Comments and Suggestions for Authors Although the statistical analysis is appropriately conducted, it’s important to clearly state that this is a cross-sectional study. This should be mentioned not only in the methods section but also in the abstract and discussion. As a result, any interpretations suggesting causality or clinical application of GDF-15 as a biomarker should be presented with caution.   the authors should explicitly highlight the novelty of their work and clearly delineate how it extends or differs from previous studies on GDF-15.   The manuscript frequently presents GDF-15 as a promising diagnostic or predictive biomarker. However, considering the cross-sectional nature of the study, such interpretations should be framed as exploratory and hypothesis-generating rather than conclusive.   Although the manuscript notes ethnic differences, its impact would be enhanced by examining potential interaction effects—for example, whether the relationship between ethnicity and outcomes varies by diabetes status, or how gender interacts with insulin resistance. Such analyses could provide more nuanced insight into demographic influences.   Some relationships, such as those involving BMI, insulin, or blood pressure, reached statistical significance but showed small effect sizes. It's important to discuss whether these findings have meaningful clinical implications     Several tables and figures lack clear labels or are not adequately referenced and interpreted in the main text. Enhancing clarity here would help readers better understand the findings.   Revise figure and table labels for greater clarity and uniformity. Correct minor formatting issues and typographical errors (e.g., incomplete email addresses). Merge repetitive content in the introduction to streamline it. In the Methods section, be more specific about how batch effects and assay calibrations were handled.

Author Response

Reply to the reviewers' report for the manuscript titled “Investigating the role of GDF-15 in diabetes and obesity: A comprehensive analysis of a cohort from the KEDP-study” submitted to Biomedicines.

We thank the reviewers for their careful reading and valuable comments and suggestions, which certainly will improve the presentation and content of the article. Find below the reply to your comments and suggestions in the same order as they appeared in your report.

Reply to reviewer 1:

Comments and Suggestions for Authors

Comment: Although the statistical analysis is appropriately conducted, it’s important to clearly state that this is a cross-sectional study. This should be mentioned not only in the methods section but also in the abstract and discussion.

Reply: Thank you for your acknowledgment of the appropriateness of the statistical analysis. We agree with you about mentioning the study design. Please see lines 39 and 339 of the revised manuscript.   

Comment: As a result, any interpretations suggesting causality or clinical application of GDF-15 as a biomarker should be presented with caution.

Reply: Thank you for the comment. We agree, and we have added the cross-sectional design as a limitation, and results should be interpreted as associations but not causation. Please see lines 389-391 of the revised manuscript.

Comment: The authors should explicitly highlight the novelty of their work and clearly delineate how it extends or differs from previous studies on GDF-15.  The manuscript frequently presents GDF-15 as a promising diagnostic or predictive biomarker. However, considering the cross-sectional nature of the study, such interpretations should be framed as exploratory and hypothesis-generating rather than conclusive.   

Reply: We thank the reviewer for this thoughtful comment. We agree that the cross-sectional design of the study limits causal inference, and we have accordingly framed our interpretations as exploratory and hypothesis-generating. In the revised manuscript, we have emphasized this point more explicitly to avoid any misinterpretation of our findings as conclusive. Please see lines 396-401 of the revised manuscript and it reads as follows:” Given the cross-sectional nature of this study, the associations observed between GDF-15 and various metabolic markers should be interpreted as exploratory and hypothesis-generating rather than indicative of causal relationships. Future longitudinal and interventional studies are warranted to validate these findings and further elucidate the potential role of GDF-15 in metabolic disease progression.”

As for the novelty of this work, the current study establishes an association between the GDF-15 marker and clinical, biochemical, and glycemic markers. For that reason, the authors stated in the discussion that the findings could have significant implications for further research into the mechanisms underlying diabetes and obesity, as well as potential therapeutic interventions aimed at modulating GDF-15 levels to mitigate metabolic risks. Furthermore, the authors stated that this study represents a substantial stride in advancing our understanding of the potential significance of GDF-15 in obesity and diabetes, highlighting its promising role as a biomarker for metabolic diseases across various ages, genders, and ethnic groups. Please see lines 342-354 and 378-391 of the revised manuscript.

Comment: Although the manuscript notes ethnic differences, its impact would be enhanced by examining potential interaction effects—for example, whether the relationship between ethnicity and outcomes varies by diabetes status, or how gender interacts with insulin resistance. Such analyses could provide more nuanced insight into demographic influences. 

Reply: Thank you for this excellent point. We have conducted an interaction between gender and insulin resistance (HOMA-IR), and this interaction was not significant, and the graphs for the male and female lines were parallel (no interaction), see the partial output below.

We further investigated the interaction between ethnicity and diabetes status, using Arab and non-diabetic individuals as the reference groups. A significant interaction was observed only among Southeast Asians (p = 0.037). We have incorporated this finding into the Discussion section of the revised manuscript. Specifically, we added the following statement on lines 387–391: “After adjusting for covariates in Table 4, a significant interaction was observed between diabetes status and ethnicity in relation to GDF-15 levels. This indicates that the association between diabetes and GDF-15 levels varies by ethnicity. Notably, diabetic Southeast Asians had significantly lower median GDF-15 levels compared to those of Arab ethnicity.”

Comment: Some relationships, such as those involving BMI, insulin, or blood pressure, reached statistical significance but showed small effect sizes. It's important to discuss whether these findings have meaningful clinical implications.

Reply: Thank you for this insightful comment. You refer to Table 3, where we reported Spearman correlations between GDF-15 and several covariates. While some of these correlations (e.g., with BMI, insulin, and blood pressure) were statistically significant, we acknowledge that the effect sizes were small (e.g., r ≈ 0.10), likely due to the large sample size. These correlations were presented to demonstrate potential associations that justified their inclusion in the subsequent quantile regression analysis. Our primary objective was not to interpret these bivariate correlations on their own, but rather to use them as a foundation for building a more robust multivariate model. We have clarified this point in the revised manuscript to avoid any misunderstanding. We added the following sentences for clarity in lines 316-322: “Although some bivariate correlations between GDF-15 and clinical variables such as BMI, insulin, and blood pressure reached statistical significance; the effect sizes were small. These findings, while statistically detectable due to the large sample size, may have limited standalone clinical relevance. However, they served as a rationale for including these covariates in the quantile regression model, which provides a more comprehensive understanding of the relationship between GDF-15 and metabolic risk factors.”

Comment: Several tables and figures lack clear labels or are not adequately referenced and interpreted in the main text. Enhancing clarity here would help readers better understand the findings.   Revise figure and table labels for greater clarity and uniformity. 

Reply: Thank you for your helpful comment. We have revised the titles and footnotes of the tables and figures to improve clarity, consistency, and interpretability. These changes follow standard reporting practices and aim to ensure that each table and figure is self-explanatory and clearly linked to the main text. We hope these revisions address your concerns.

Comment: Correct minor formatting issues and typographical errors (e.g., incomplete email addresses). Merge repetitive content in the introduction to streamline it. In the Methods section, be more specific about how batch effects and assay calibrations were handled.

Reply: Thank you for your comment. The authors have conducted a thorough reading of the revised manuscript and fixed some typos. We hope these revisions meet your expectations and improve the clarity and quality of the manuscript.

We sincerely thank the reviewer for the thoughtful feedback, which helped improve this manuscript. We trust the revised version now meets the journal’s standards and look forward to your favorable response.

Reviewer 2 Report

Comments and Suggestions for Authors

The manuscript details an interesting topic that highlights the role of GDF-15 in metabolic conditions. The methodology is appropriate and replicable. However, the following conclusion is an overstatement: "These findings substantially enhance our understanding of GDF-15's prospective role as a marker of diabetes and obesity amongst specific genders, age groups, and ethnicities."

The findings suggest that GDF-15 represents a state of heightened inflammation that varies according to several parameters but is not yet a confirmed marker of either diabetes or obesity.

What would be more interesting is the temporal relationship of GDF-15 with diabetes and obesity. Do the levels improve with disease remission?

Author Response

Reply to the reviewers' report for the manuscript titled “Investigating the role of GDF-15 in diabetes and obesity: A comprehensive analysis of a cohort from the KEDP-study” submitted to Biomedicines.

We thank the reviewers for their careful reading and valuable comments and suggestions, which certainly will improve the presentation and content of the article. Find below the reply to your comments and suggestions in the same order as they appeared in your report.

Reply to Reviewer 2:

Comments and Suggestions for Authors

Comment: The manuscript details an interesting topic that highlights the role of GDF-15 in metabolic conditions. The methodology is appropriate and replicable.

Reply: Thank you for your positive feedback. We appreciate your recognition of the relevance of the topic and the strength of the methodology.

Comment: However, the following conclusion is an overstatement: "These findings substantially enhance our understanding of GDF-15's prospective role as a marker of diabetes and obesity amongst specific genders, age groups, and ethnicities."

Reply: Thank you for this valuable comment. Our intention was to convey that our findings contribute to the growing body of literature on the potential role of GDF-15 as a biomarker for diabetes and obesity across different genders, age groups, and ethnicities. In response to your suggestion, we have removed the word “substantially” to avoid overstatement and ensure a more balanced interpretation. We hope this revision addresses your concern.

Comment: The findings suggest that GDF-15 represents a state of heightened inflammation that varies according to several parameters but is not yet a confirmed marker of either diabetes or obesity.

Reply: Thank you for your thoughtful comment. We agree that GDF-15 should not yet be considered a confirmed diagnostic marker for diabetes or obesity. However, numerous studies in the literature have reported elevated GDF-15 levels in individuals with type 2 diabetes and its association with obesity and other metabolic disturbances. Our study adds to this growing body of evidence by further exploring these associations in a diverse cohort.

Comment: What would be more interesting is the temporal relationship of GDF-15 with diabetes and obesity. Do the levels improve with disease remission?

Reply: Thank you for this insightful comment. As noted in the manuscript, the current study is cross-sectional in design, which limits our ability to assess temporal relationships or changes in GDF-15 levels over time. Longitudinal data would indeed be required to determine whether GDF-15 levels improve with disease remission. We agree that this is an important area for future research and would make for a valuable follow-up study. The following sentences were added in lines 396-301 of the revised manuscript: “Given the cross-sectional nature of this study, the associations observed between GDF-15 and various metabolic markers should be interpreted as exploratory and hypothesis-generating rather than indicative of causal relationships. Future longitudinal and interventional studies are warranted to validate these findings and further elucidate the potential role of GDF-15 in metabolic disease progression.”

We sincerely thank the reviewer for the thoughtful feedback, which helped improve this manuscript. We trust the revised version now meets the journal’s standards and look forward to your favorable response.

Reviewer 3 Report

Comments and Suggestions for Authors

A member of the TGF-β/BMP superfamily, growth differentiation factor 15 (GDF15) is a stress-responsive cytokine that is released into the bloodstream following tissue injury, hypoxia, and a pro-inflammatory response. In addition to macrophages, vascular smooth muscle cells, cardiomyocytes, adipocytes, and endotheliocytes are also the source of GDF15. GDF15 levels in blood serum are known to be elevated in type 2 diabetes mellitus and are linked to FPG levels. Furthermore, it was found that gestational diabetes mellitus was associated with higher GDF15 levels. Obese patients showed an increase in GDF15 levels in their blood serum, which could be a sign of a poor prognosis following heart surgery. Therefore, GDF15 needs more study, including in specific populations, as it can serve as a biomarker of a number of clinical disorders in the human body.

In theory, the paper tackles a pressing issue by evaluating the function of GDF15 in the nation's general population and the potential for it to serve as a biomarker for diabetes and obesity. It is provided in the conventional format characteristic of the original research.
In half of the cases, the authors drew on 2020–2025 literature when writing the paper.

The study's authors' hypothesis essentially states that it is necessary to evaluate the GDF15 level in the Kuwaiti population while accounting for factors including gender, age, nationality, and potential associations with obesity and diabetes. It is appropriate to test hypotheses using the suggested research methodology.
It is understandable that third parties cannot replicate this study because it was carried out on a Kuwaiti resident; nevertheless, it can be carried out on people in other nations.

Any article illustration is, in theory, suitable and enables you to present research findings in a clear and succinct manner. The key is that the illustrated content should be well-designed so that the reader can grasp the essential points of the study and its findings without having to read the remainder of the article.
Although the authors have provided a detailed explanation of their statistical analysis methods, several of the findings of their research cannot be appropriately interpreted by the authors (see the comments).

Notes: 1. The text was written with footnotes for citations in lines 75 and 78; 2. The sentence in line 138 is missing a period; 3. Table 1: The annotation should explain the full abbreviation; 4. The authors of the GDF15 levels study state that they have found statistically significant differences between the groups, but I apologize for the wide range of values (the standard deviation overlaps and there is no question of statistical significance even when converted to the error of the average, which is essentially used in statistical programs) (Figures 1 and 2). Here, the median and the interquartile range are more appropriate; 5. Since some of the authors' values in Table 2 have increased, it is preferable to create a space between the groups. To make it obvious to the reader, the groups in each position might be numbered to show which group is assigned the significance of the differences.
6. The correlation analysis parts need to be rewritten. Regarding the assessment of correlation values for both Pearson and Spearman, the authors lack information.

These values have varying degrees of significance:
A very low association is between 0.0 and 0.2; a weak association is between 0.3 and 0.4; Medium-strength association: 0.5–0.7; 0.8–1.0 is a strong relationship.

Therefore, the direct and inverse connections that are found below 0.4 aprory are deemed to be negligible. There are occasionally expectations that the relationship will become stronger as the number of observations rises, but in this instance, when the groups are big, such expectations have perished.

It is impossible to assess the conclusions' accuracy based on the remarks.

Yes, all moral principles have been followed.

Author Response

Reply to the reviewers' report for the manuscript titled “Investigating the role of GDF-15 in diabetes and obesity: A comprehensive analysis of a cohort from the KEDP-study” submitted to Biomedicines.

We thank the reviewers for their careful reading and valuable comments and suggestions, which certainly will improve the presentation and content of the article. Find below the reply to your comments and suggestions in the same order as they appeared in your report.

Reply to Reviewer 3:

Comments and Suggestions for Authors

A member of the TGF-β/BMP superfamily, growth differentiation factor 15 (GDF15) is a stress-responsive cytokine that is released into the bloodstream following tissue injury, hypoxia, and a pro-inflammatory response. In addition to macrophages, vascular smooth muscle cells, cardiomyocytes, adipocytes, and endotheliocytes are also the source of GDF15. GDF15 levels in blood serum are known to be elevated in type 2 diabetes mellitus and are linked to FPG levels. Furthermore, it was found that gestational diabetes mellitus was associated with higher GDF15 levels. Obese patients showed an increase in GDF15 levels in their blood serum, which could be a sign of a poor prognosis following heart surgery. Therefore, GDF15 needs more study, including in specific populations, as it can serve as a biomarker of a number of clinical disorders in the human body.

Comment: In theory, the paper tackles a pressing issue by evaluating the function of GDF15 in the nation's general population and the potential for it to serve as a biomarker for diabetes and obesity. It is provided in the conventional format characteristic of the original research. In half of the cases, the authors drew on 2020–2025 literature when writing the paper.

Reply: Thank you for acknowledging the relevance of our work on GDF-15 as a potential biomarker for diabetes and obesity in the general population. We agree that this is a timely and important area of investigation. We also appreciate your feedback regarding the use of recent literature and have ensured that the references cited are up to date and reflective of current research (2020–2025) to support the study's context and findings.

Comment: The study's authors' hypothesis essentially states that it is necessary to evaluate the GDF15 level in the Kuwaiti population while accounting for factors including gender, age, nationality, and potential associations with obesity and diabetes. It is appropriate to test hypotheses using the suggested research methodology.
It is understandable that third parties cannot replicate this study because it was carried out on a Kuwaiti resident; nevertheless, it can be carried out on people in other nations.

Reply: Thank you for your thoughtful comment. We agree that evaluating GDF-15 levels in the Kuwaiti population—while accounting for factors such as gender, age, and nationality—offers important insights into population-specific patterns and associations with metabolic conditions. Although the study was conducted on residents of Kuwait, we believe the methodology is robust and replicable, and the findings may offer valuable reference points for similar studies in other populations. We hope our work encourages further cross-population comparisons that could advance the global understanding of GDF-15’s role in metabolic health.

Comment: Notes: 1. The text was written with footnotes for citations in lines 75 and 78;

Reply: Thank you for the observation, we used footnotes to illustrate some symbols or the type of test conducted.

Comment: 2. The sentence in line 138 is missing a period.

Reply: It is corrected.

Comment: 3. Table 1: The annotation should explain the full abbreviation

Reply: Thank you for your comment. We confirm that all abbreviations used in Table 1 are explained in full in the corresponding footnotes below the table.

Comment: 4. The authors of the GDF15 levels study state that they have found statistically significant differences between the groups, but I apologize for the wide range of values (the standard deviation overlaps and there is no question of statistical significance even when converted to the error of the average, which is essentially used in statistical programs) (Figures 1 and 2). Here, the median and the interquartile range are more appropriate.

Reply: Thank you for this valuable observation. As noted in the Methods section, we used non-parametric tests (Mann–Whitney or Kruskal–Wallis) when the assumption of normality was not met, and parametric tests (t-tests or ANOVA) when normality was satisfied. In cases of skewed distributions, we reported the median and interquartile range (IQR), as appropriate. Figures 1 and 2 have been reviewed to ensure consistency between the statistical methods used and the data presentation.

Comment: 5. Since some of the authors' values in Table 2 have increased, it is preferable to create a space between the groups. To make it obvious to the reader, the groups in each position might be numbered to show which group is assigned the significance of the differences.

Reply: Thank you for your comment. We agree that clearer spacing and formatting would improve readability. We believe this will be addressed during the journal’s typesetting and formatting stage, where table alignment and layout are standardized. However, we have reviewed Table 2 to ensure that group comparisons and significance markers are clearly labeled and interpretable.

Comment: 6. The correlation analysis parts need to be rewritten. Regarding the assessment of correlation values for both Pearson and Spearman, the authors lack information.

These values have varying degrees of significance:
A very low association is between 0.0 and 0.2; a weak association is between 0.3 and 0.4; Medium-strength association: 0.5–0.7; 0.8–1.0 is a strong relationship. Therefore, the direct and inverse connections that are found below 0.4 aprory are deemed to be negligible. There are occasionally expectations that the relationship will become stronger as the number of observations rises, but in this instance, when the groups are big, such expectations have perished. It is impossible to assess the conclusions' accuracy based on the remarks.

Reply: Thank you for this detailed comment. We stated earlier to the other reviewer that we are presenting only Spearman correlation (due to the presence of outliers) in an attempt to gauge the strength of the linear relationship (correlation) and thus use this as a springboard to build the quantile regression model. So, you can think of the correlation as an intermediate step to identify those covariates with a strong correlation. We hope that you are OK with that.

Comment: Yes, all moral principles have been followed.

Reply: Thank you for your kind acknowledgment.

We sincerely thank the reviewer for the thoughtful feedback, which helped improve this manuscript. We trust the revised version now meets the journal’s standards and look forward to your favorable response.

Round 2

Reviewer 1 Report

Comments and Suggestions for Authors

Dear Authors,

Thank you for your thoughtful and detailed responses to the reviewers’ comments and for submitting a thoroughly revised version of your manuscript entitled “Investigating the role of GDF-15 in diabetes and obesity: A comprehensive analysis of a cohort from the KDEP-study.”

I appreciate the effort you have put into addressing each point raised during the initial review. It is clear that you have taken the feedback seriously, and the revisions you have made have strengthened the manuscript considerably. In particular, I found your clarifications regarding the cross-sectional nature of the study, the revised discussion around causality and clinical implications, and the additional interaction analyses to be appropriate and well-implemented. The improved clarity in tables, figures, and formatting also enhanced the readability of the manuscript.

Overall, your responses were comprehensive and convincing.

Kind regards

Reviewer 3 Report

Comments and Suggestions for Authors

Growth differentiation factor 15 (GDF15), a cytokine that is sensitive to stress and is a member of the TGF-β/BMP superfamily, is implicated in metabolic syndrome, which includes obesity and diabetes mellitus. To find potential associations between its serum levels and pathology, the level of GDF15 should definitely be assessed in different human populations since it can function as a biomarker of pathogenic processes in the human body. Given that the majority of the cited material is no more than 2020, this makes the current work pertinent. The work's primary goal was to evaluate the GDF15 level in various ethnic groups while accounting for Kuwaiti inhabitants' gender characteristics, which can be extrapolated to other nationalities. The authors enhanced the article's quality by making a number of changes.

But when you look at Figures 1 and 2, you automatically assume that the mean and standard deviation are how the results are displayed. Whether the data are displayed graphically as the mean and standard deviation or as the median and upper and lower quartiles, this should be indicated in the figures' annotations. since recommended by the Statistica statistical software package or Excell, it is preferable to supply the upper and lower quartiles and construct figures in the form of hanging boxes, since this offers a more aesthetically pleasing display of the data. This is a personal opinion, though, and the writers may decide to disregard it.

I have no ethical concerns regarding the amended version, just as I had none when I reviewed the work in the first round.